# Oxidative Stress in Cancer Immunotherapy: Molecular Mechanisms and Potential Applications

**DOI:** 10.3390/antiox11050853

**Published:** 2022-04-27

**Authors:** Ruolan Liu, Liyuan Peng, Li Zhou, Zhao Huang, Chengwei Zhou, Canhua Huang

**Affiliations:** 1School of Basic Medical Sciences, Chengdu University of Traditional Chinese Medicine, Chengdu 611137, China; 2020ks012@stu.cdutcm.edu.cn; 2State Key Laboratory of Biotherapy and Cancer Center, West China Hospital, and West China School of Basic Medical Sciences & Forensic Medicine, Sichuan University, and Collaborative Innovation Center for Biotherapy, Chengdu 610041, China; 2019324060051@stu.scu.edu.cn (L.P.); 2015224060079@stu.scu.edu.cn (L.Z.); huangzhao@scu.edu.cn (Z.H.); 3Department of Thoracic Surgery, The Affiliated Hospital of Ningbo University School of Medicine, Ningbo 315020, China

**Keywords:** reactive oxygen species (ROS), oxidative stress, antitumor immune response, cancer immunotherapy

## Abstract

Immunotherapy is an effective treatment option that revolutionizes the management of various cancers. Nevertheless, only a subset of patients receiving immunotherapy exhibit durable responses. Recently, numerous studies have shown that oxidative stress induced by reactive oxygen species (ROS) plays essential regulatory roles in the tumor immune response, thus regulating immunotherapeutic effects. Specifically, studies have revealed key roles of ROS in promoting the release of tumor-associated antigens, manipulating antigen presentation and recognition, regulating immune cell phenotypic differentiation, increasing immune cell tumor infiltration, preventing immune escape and diminishing immune suppression. In the present study, we briefly summarize the main classes of cancer immunotherapeutic strategies and discuss the interplay between oxidative stress and anticancer immunity, with an emphasis on the molecular mechanisms underlying the oxidative stress-regulated treatment response to cancer immunotherapy. Moreover, we highlight the therapeutic opportunities of manipulating oxidative stress to improve the antitumor immune response, which may improve the clinical outcome.

## 1. Introduction

Generally, the traditional treatments of cancer involve surgery, chemotherapy, radiotherapy, or molecular targeted therapy. However, even the best regimens within these strategies rarely cure late-stage tumors [1]. At the turn of the last century, researchers started to investigate the possibility of taking advantage of the patient’s immune system to clear cancer cells, which is now thought to be the greatest chance for cancer treatment [1]. In the 1950s, Thomas and Burnet proposed the immune surveillance theory that the immune system eliminates cancer cells in the body by recognizing tumor-specific antigens [2,3]. However, malignant tumors also initiate and develop in people with fully functional host immune systems [4], indicating that the immune surveillance theory is not all-inclusive. Subsequently, the theory of immunoediting was proposed, which describes that tumor development consists of an elimination phase, equilibrium phase and escape phase (Figure 1), leading to the tumor evasion of host immune surveillance [5].

To enhance the antitumor effect, cancer immunotherapies have been developed based on the above theories. However, only a small number of patients benefit from these immunotherapeutic strategies, largely due to the immunosuppressive mechanisms developed by tumor cells. For example, once antigens on the surface of tumor cells are recognized, the tumor cells start to overexpress suppressive cytokines, such as IL6, IL10 or TGF-β, which can block the immune system [6]. Moreover, cancer cells also secrete immunosuppressive molecules, such as PGE2, which bind specifically to EP2 and EP4 receptors on cytotoxic T lymphocytes (CTLs), inhibiting CTLs’ survival and normal functioning. In addition, tumor cells can recruit suppressive cells, such as regulatory T cells (Tregs) or suppressive cytokines, including IL10. This phenomenon is even amplified by transformed fibroblasts surrounding the tumor, which release chemokine ligand 2 (CCL2) and the chemokine (C-X-C motif) ligand 2 (CXCL2) to enhance recruitment. Emerging evidence suggests that these responses of cancer cells are intimately correlated with ROS-induced oxidative stress. In detail, ROS can regulate a wide variety of tumor immune-response processes, such as the release and presentation of tumor antigens, stimulation of immune effector cells and inhibition of immune regulatory cells [7]. Therefore, it is essential to summarize the recent progress regarding the effects of ROS-induced oxidative stress in cancer therapy.

In this review, we present a summary of several types of cancer immunotherapy strategies. We further discuss the molecular mechanistic basis of the interplay between oxidative stress and anticancer immunity, with an emphasis on oxidative stress-regulated treatment response to cancer immunotherapy. On this basis, we highlight the therapeutic opportunities of manipulating oxidative stress to improve the antitumor immune response, which may be helpful to the development of more efficient clinical management strategies based on immunotherapy.

## 2. Overview of the Cancer Immunotherapy

Cancer immunotherapies have been developed based on studies of tumor escape mechanisms, which could be categorized as either active or passive immunotherapies [8,9]. Active immunotherapies are suitable for an immune-responsive host and/or an immunogenic tumor, because it stimulates and enhances the host’s anti-tumor immune response. The commonly used active immunotherapies include immune checkpoint inhibitors (ICIs), cancer vaccines (CVs) and cytokine treatment. Passive immunotherapies, such as monoclonal antibodies, oncolytic viruses (OVs) and adoptive cell therapy (ACT), treat tumors by injecting effector cells or antibody conjugates that can directly kill tumor cells, independent of the host’s immune function status [10] (Figure 2).

### 2.1. Active Immunotherapies

***Immune checkpoint inhibitors (ICIs).*** Immune checkpoint inhibitors (ICIs) are one of the most broadly successful cancer immunotherapies to date [11]. Physiologically, immune checkpoints maintain appropriate immune responses by preventing excessive T-cell activation, thus protecting healthy tissues from immune attack. However, by harnessing this mechanism, tumor cells upregulate immune checkpoint proteins, thus escaping from the immune system. Therefore, ICIs have been developed to block the binding of checkpoint proteins to their partner proteins, preventing the inhibitory signal and recovering antitumor immunity [12]. The most common ICIs are PD-1/PD-L1 blockade and CTLA-4 inhibitors [13]. The receptor PD-1 is usually expressed on the surface of activated T cells, enabling the recognition of cancer cells [14,15]. To avoid T-cell-induced elimination, tumor cells express PD-L1, which binds to and inactivates PD-1 [14]. Thus, disrupting the binding of PD-1 and PD-L1 could induce tumor cell death [16]. Another immune checkpoint, CTLA4, is a co-inhibitory molecule regulating T-cell activation. The interaction of CD80/86 and CTLA-4 inhibits T-cell activation, thus promoting tumor progression [13]. Ipilimumab, which targets CTLA4, is the first checkpoint inhibitor approved for the treatment of unresectable or metastatic melanomas [17].

***Cancer vaccines (CVs).*** Cancer vaccines are subclassified as either prophylactic or therapeutic interventions, which boost the immune system to clear cancer cells. Prophylactic vaccines aim to reduce cancer incidence, morbidity and mortality. Currently, several vaccines available in clinical practice can efficiently prevent tumor progression. For instance, prophylactic cancer vaccines have been successfully used for hepatocellular carcinoma, secondary to hepatitis B virus (HBV), and squamous cell carcinoma, secondary to human papillomavirus (HPV) [18]. It is widely accepted that infection with high-risk HPV (such as HPV16 and HPV18) is a major risk factor for cervical cancer [19]. Some research data indicated that both of the two approved HPV vaccines (the bivalent vaccine for HPV16 and HPV18, and the quadrivalent vaccine for HPV6, HPV11, HPV16 and HPV18) have efficacies of 90–100% [20]. Currently, therapeutic tumor vaccines include tumor cell vaccines, long peptide vaccines and gene vaccines [21]. In 2010, the only FDA-approved vaccine was sipuleucel-T for prostate cancer treatment [22]. In addition, a personalized vaccine was proposed, which was selected according to the own human leukocyte antigen -A (HLA-A) type of patients and pre-existing immune memory to achieve the goal of personalized therapy [21,23].

***Cytokine treatment.*** Cytokines are small proteins secreted by cells to regulate the innate and adaptive immune systems. Cytokine treatment is the first active immune therapy to be applied in the clinic, in which cytokines are directly injected to boost the immune system [24]. IL-2 is a powerful T-cell growth factor, which can stimulate the activity and proliferation of CD4^+^ and CD8^+^ T cells. Currently, IL-12 is used for the treatment of various types of cancers [25,26]. Additionally, interferons normally trigger immune responses through inducing the maturation of immune cells, such as NK cells and macrophages [27].

### 2.2. Passive Immunotherapies

***Monoclonal antibodies.*** The monoclonal antibody was the first validated treatment, aside from classical therapies. Naked antibodies are one type of therapeutic strategy that can activate apoptosis and can be used to directly trigger ADCC via NK cells. Another type is the conjugated antibody, which carries an active factor, such as a radioisotope, a toxin or a specific drug. In this case, the antibody is used as a targeting tool to deliver the effector molecule directly to the tumor [28] (Figure 3a).

***Oncolytic viruses (OVs).*** OVs are viruses that exist in nature or are genetically engineered to selectively replicate in tumor cells to activate immune responses and induce tumor cell lysis. Currently, many OVs are in clinical trials, most of which belong to DNA viruses, such as Herpeviridae and Adenoviridae [29], and a small portion of which are RNA viruses. Different from gene therapy using viruses as gene delivery vectors, OVs themselves can act as active drugs [30]. OVs exert antitumor activity, mainly through the following ways: the selective replication in cancer cells, resulting in tumor lysis; cleavage of the released tumor-associated antigen to activate an immune response, thereby eliminating cancer cells; and induction of cytokine release from tumor cells, which, in turn, eliminates metastatic tumors [31]. Reovirus, a naturally occurring oncolytic virus isolated from the respiratory tract or intestinal tract of humans, has a natural preference for cancer cells. In addition, serotype 3 reovirus has been found to have oncolytic activity in multiple tumor types [32]. Research has shown that reovirus can also stimulate the body’s own immune system to kill liver cancer cells and clear hepatitis C virus (HCV), a common virus that causes liver cancer [33]. Moreover, oncolytic viruses can also be genetically modified to include therapeutic genes, cytokines, such as granulocyte-macrophage colony-stimulating factor (GM-CSF) and IL-12, and play a synergistic role in killing tumor cells through multiple pathways, which can effectively avoid the current single-target anticancer drugs. GM-CSF, a glycoprotein, can promote the proliferation and maturation of neutrophils to improve their anti-tumor and anti-infection capabilities. Many data show that ROS are associated with the pathogenesis and the development of Crohn’s disease [34,35]. A study evaluating the intracellular redox status of neutrophils in patients with CD reported that GM-CSF can induce excessive ROS production by human neutrophils [34]. Talimogene laherparepvec (T-VEC)-modified herpes simplex virus type 1 (HSV-1) with GM-CSF in its genome, is the first OV approved for melanoma treatment [30] (Figure 3b).

***Adoptive cell therapy (ACT).*** Engineered cells for ACT include T-cell receptor T cells (TCR-Ts), chimeric antigen receptor T cells (CAR-Ts) and CAR-NK cells. These cells are engineered to express fusion proteins on the cell surface, whose extracellular fragment is a monoclonal antibody-like receptor that exhibits high specificity to recognize the full antigen on a tumor cell [36]. The earliest effective ACT was used to treat hematologic malignancies by bone marrow transplantation [37]. CAR-T cell therapy was firstly described in 1993. In detail, T cells collected from patient were genetically engineered to express CARs specific for recognizing the antigen present on tumor cells, then readministered to the patients. Currently, the most common targets of CAR-T therapy are B-cell surface antigens, such as the extracellular glycoproteins CD19 and CD20 [38]. Thus, CAR-T therapy has yielded unprecedented efficacy in B-cell malignancies. Unlike MHC-independent CARs, TCRs respond to tumor-associated intracellular antigens presented by MHCs [39]. In another type of ACT, cells were collected from patients’ peripheral blood or tumor in situ, amplified and activated in vitro, and then transfused back into patients for antitumor therapy [40]. This kind of ACT involves tumor-infiltrating immune cell (TIL) therapy, cytokine-induced killing (CIK) cell therapy, lymphocyte cytokine-activated killing (LAK) cell therapy and NK-cell therapy (Figure 3c).

## 3. Mechanistic Basis of ROS in Cancer Immunotherapy

### 3.1. ROS Generation and Elimination

ROS are toxic intermediates or byproducts of aerobic metabolism produced by almost all organisms and cells. This type of oxygen-derived molecule includes radicals, as exemplified by hydroxyl radicals (•OH) and superoxide anions (O^2•−^), and nonradical species, such as hydrogen peroxide (H_2_O_2_) [41]. Endogenous ROS are primarily generated in mitochondria, the endoplasmic reticulum (ER) and peroxisomes [42]. The two major sources of intracellular ROS are the respiratory electron transport chain (ETC) of mitochondria [43,44,45] and oxidoreductase enzymes, such as cyclooxygenases (COXs) and NADPH oxidases (NOXs) [45,46]. In addition, ROS can typically arise from exogenous stimulation, including ultraviolet radiation, heavy metals [47], cancer chemotherapy and radiotherapy.

Typically, ROS act as signaling molecules in multiple cellular physiological processes at moderate levels. However, excessive ROS induce oxidative damage to biomolecules, including protein oxidation, lipid peroxidation and DNA damage [48]. To counteract the deleterious effects of overloaded ROS, organisms have evolved some antioxidant defense mechanisms that can be divided into direct-acting endogenous and exogenous systems. The primary endogenous systems are some antioxidant enzymes, including superoxide dismutases (SODs) and catalase (CAT) [49]. The SODs family contains three metalloenzymes that catalyze the same reaction in different regions of the cell. The activity of SODs is closely related to the degree of oxidative damage. CAT is mainly localized in peroxisomes and catalyzes the conversion of H_2_O_2_ into H_2_O and O_2_ [50]. Specifically, SODs catalyze the dismutation of O^2•−^ to H_2_O_2_, which is decomposed into H_2_O and O_2_ through CAT as well as glutathione peroxidases (GPXs) [51]. Moreover, GSH is the primary non-enzymatic antioxidant in living cells and is abundant in various tumor types [52]. Under normal physiological conditions, GSH is mainly in its reduced form, maintaining a reducing environment in subcellular compartments, such as the mitochondria, nucleus and cytosol. However, under oxidative stress, GSH can be directly converted to glutathione disulfide (GSSG) by reacting with ROS or indirectly converted to GSSG by a reaction catalyzed GPXs [53]. In addition to these endogenous synthesis enzymes and small molecules, some diets provide exogenous defenses, including nutrients, such as ascorbic acid, lecithin oil, β-carotene and vitamin E [54,55]. Furthermore, the antioxidant systems include heme oxygenase isoenzymes (HO-1/2), carbonyl reductases (CBRs) and Sirtuin 3 (SIRT3) [56].

### 3.2. Regulatory Role of ROS in the Immune System

The immune system contains innate and adaptive immunity, both playing essential roles in the maintenance of organism integrity by protecting against invading pathogens and for the elimination of transformed cells [57]. Innate immune cells, composed of NK cells, macrophages and neutrophils cells, participate in tumor suppression, either by directly killing tumor cells or triggering an adaptive immune response [58]. T cells and B cells are pivotal components of the adaptive immune system, among which T cells are responsible for cell-mediated immune responses, whereas B cells are involved in humoral immune responses. Immune cells can recognize and eliminate tumor cells, while tumor cells can evade immunosurveillance by subverting antitumor immune responses [2]. Moreover, both immune cells and tumor cells are influenced by the tumor microenvironment with persistently high ROS levels. Compared with the surrounding normal tissues, tumors often exhibit higher ROS levels [59]. Specifically, ROS can function as a direct chemical attractant to regulate immune-cell recruitment. In addition, immune cells can recognize pathogen-associated molecular patterns (PAMPs), and damage-associated molecular patterns (DAMPs) trigger intracellular signaling events by resulting in elevated ROS production and proinflammatory cytokine secretion [60]. Substantial evidence indicates that ROS-induced oxidative stress is closely related to multifaceted aspects of the immune response. Here, we summarize the regulatory role of ROS in different immune cells, including T cells, APCs, Tregs, myeloid-derived suppressor cells (MDSCs) and tumor-associated macrophages (TAMs). In addition, we also focus on the molecular mechanisms underlying oxidative stress-regulated treatment response to cancer immunotherapy (Figure 4).

#### 3.2.1. ROS and T Cells

Currently, many immunotherapies (such as ACT, ICIs and TCV) targeting or harnessing T cells have demonstrated therapeutic efficacy for a broad range of human malignancies. T cells, especially CTLs, are lymphocytes possessing specific cell-killing activity [61,62]. Studies have found that tumors infiltrated with more CTLs are more likely to respond to immunotherapy [62,63]. T-cell activation is dependent on three signals: TCR stimulation, co-stimulation and cytokine release. T cells express specific receptors for antigens called TCRs on their cell surface. The antigen–MHC molecule complexes are presented as APCs, such as dendritic cells, B cells and macrophages. TCR signaling is initiated by T-cell recognition of the MHC-associated peptide antigen on APCs. However, it requires additional costimulatory signals to fully activate T cells, since MHC-peptide recognition is usually not sufficient. CD28, the most principal costimulatory molecule expressed on T cells, interacts with the B7 ligands CD80 (B7-1) and CD86 (B7-2) on APCs to enhance T-cell activation and cytokine release [64]. The checkpoint inhibitor CTLA-4 competitively binds to CD80/CD86 with a higher affinity and limits T-cell responses [65]. Additionally, a large quantity of evidence demonstrates that cytokines play distinct roles at different stages of T-cell homeostasis [66]. More intriguingly, some studies indicate that T cells with increased expressions of antioxidant enzymes and sulfhydryl groups on the surface exhibit longer survival and greater antitumor activity [67,68], suggesting that an enhanced antioxidant capacity may facilitate the antitumor efficacy of T cells. To date, increasing evidence has shown that ROS affect T-cell activation, proliferation and survival, indicating that modulating ROS in the immune microenvironment can enhance antitumor immune responses [69,70] (Table 1).

***T cell activation.*** Moderate levels of ROS participate in the activation, proliferation and differentiation of T cells, whereas a high level of ROS is the main factor for T-cell inhibition. Interestingly, it was previously reported that, after a reduction in mitochondrial ROS, T cells could not be consistently activated even under CD28 or CD3 stimulation [85]. In addition, T-cell activation requires the induction of the MAPK signaling pathway to stimulate TCR. Studies have shown that H_2_O_2_ in mitochondria can amplify the MAPK signaling pathway when stimulated by antigens, leading to T-cell activation and proliferation [86]. Moreover, ROS can activate the mammalian target of rapamycin (mTOR) and AMP-activated protein kinase (AMPK), causing the peroxisome proliferator-activated receptor-γ (PPAR-γ) and PPAR-γ coactivator 1-α (PGC-1α) cascade, which promotes the activation of CD8^+^ T cells [87]. The ROS-induced redox state of kinases leads to the activation of the transcription factor nuclear factor of activated T cells (NFATs) and induces IL-2 production to promote the proliferation of activated T cells [88,89]. Specifically, the enhanced production of ROS might contribute to mitochondrial injury, since it promotes Ca^2+^ release [88]. The increase in intracellular Ca^2+^ levels results in NFAT activation that induces IL-2 gene expression and subsequent signaling events, resulting in T-cell proliferation [90]. However, multiple evidence has also revealed that high ROS levels in the TME could impair the activation, proliferation and antitumor function of T cells [91,92]. Thus, the efficacy of adoptive T-cell immunotherapy may be enhanced through decreasing ROS levels. For example, the immunosuppressive receptor PD-1 is primarily expressed in activated T cells. Once activated by the PD-L1 ligand, PD-1 can exert negative immunoregulatory effects [93]. Another case has shown that, in mouse models, sensitivity to PD-1 blockade relies on the cellular ability of oxygen consumption and ROS levels in the tumor environments [87,94]. Therefore, these studies illustrate that ROS play multifaceted roles in regulating T-cell activation, which further affects the efficacy of immunotherapy.

***T cell differentiation.*** It is well known that ROS levels increase after TCR signaling-induced T-cell activation [95]. As expected, ROS also regulate the differentiation of various T-cell subsets. For instance, higher levels of ROS in the environment tend to favor a Th2 immune phenotype [96]. In contrast, lower levels of ROS are associated with disease-causing Th1 and Th17 cell differentiation [77] (Figure 5). Moreover, H_2_O_2_-mediated TCR activation promotes mitochondrial ROS production, which enhances FasL expression in T cells and contributes to T-cell activation-induced cell death (AICD) [97].

#### 3.2.2. ROS and APCs

***Dendritic cells.*** DCs are professional antigen-presenting cells essential in the antigen-specific immune response. Immature DCs have a strong migratory ability, while mature DCs can capture various antigens and present them to T cells to initiate an antigen-specific immune response [98]. ROS can trigger the differentiation of DCs from monocyte precursors or hematopoietic cells [99]. However, excessive ROS inhibits the ability of DCs to present local antigens to T cells by inducing a chronic activation of the endoplasmic reticulum stress response [100]. Studies have shown that, in the process of DC differentiation induced by GM-CSF, DCs, at the early stage of differentiation, have lower ROS levels than those at the later stage, but they are more efficient in stimulating T cells [101]. These aforementioned effects can impede effective antitumor immune responses. Moreover, cathepsins (Cts) mainly exist in lysosomes and play a proteolytic role in a physiological state. Recent studies have shown that cathepsin expression levels are up-regulated in a variety of human tumors, especially Cathepsin B [102,103]. ROS can inhibit cathepsin hydrolysis in DCs, thus inhibiting antigen processing. In contrast, there are also studies suggesting a positive effect of ROS on DCs. ROS play a critical role in the initiation of DC-mediated antitumor reactions induced by the interferon-stimulating factor (STING) and SUMO-specific protease 3 (SENP3) [104]. Low levels of ROS can promote the transmission of cytoplasmic antigens by dendritic cells through lysosomal escape and antigen protection, resulting in efficient cross-presentation of antigens and a strong CD8+ T-cell response [105]. In addition, studies have shown that ROS also promote the maturation of antigen-presenting cells by activating several signaling pathways, including ERK, NF-κB and mTOR [100].

***B cells.*** B cells are derived from bone marrow pluripotent stem cells and are specialized antigen-presenting cells. There are various membrane surface molecules on the surface of B cells that mediate humoral immune responses by producing antibodies, such as leukocyte differentiation antigen, MHC and various membrane surface receptors [106]. In general, ROS may be involved in several stages of B-cell development, including activation, differentiation and death. A study found that ROS can mediate the B-cell receptor (BCR)-induced activation of B cells by activating the NF-κB and PI3K signaling pathways [107]. The elevated level of ROS in the microenvironment can promote the expression of HIF-1α and nuclear factor erythroid 2-related factor 2 (Nrf2), thereby regulating multiple stages of B-cell development [108]. ROS have also been found to determine a series of activities following B-cell activation. For example, low levels of ROS can induce the differentiation of B cells into plasma cells [109]. In addition, ROS can regulate apoptosis and autophagy in B cells [110]. For example, studies have found that the p66SHC protein can affect B cells in different ways. Moreover, it can promote B-cell death through autophagy by increasing ROS production, and antagonize BCR survival signals to promote cell apoptosis [111].

#### 3.2.3. ROS and Other Immunosuppressive Cells in the TME

Some immune cells can maintain peripheral tolerance and suppress antigen-specific immune responses in anticancer immunity. Tumor-associated macrophages (TAMs), myeloid-derived suppressor cells (MDSCs) and Tregs can produce excessive ROS, leading to a resistance to tumor immunotherapy through the apoptosis-related factor ligand pathway.

***Tumor-associated macrophages.*** TAMs are the most abundant immune-related cells in the TME, which is closely related to tumor development and metastasis [112]. TAMs could be subclassified into the immune-stimulatory macrophages (M1) and immune-regulatory macrophages (M2) subtypes. The reprograming or repolarization of TAMs to an antitumor phenotype may be an effective strategy to enhance the efficacy of immunotherapy [113]. Through transforming tumor-stimulating M2-like TAMs into M1-like antitumor phenotypes, TAMs can be used as effective cancer therapies [114]. Additionally, a steady increase in TME ROS levels may promote the differentiation of TAM into M2 subtypes [115]. In the co-culture model of neuroblasts and macrophages, the authors found increased lipid biosynthesis in TAMs, leading to the enrichment of phospholipids and sphingomyelin. Compared with the controls, tumor-induced macrophages are more capable of producing ROS when stimulated by these metabolic changes [116]. TAMs isolated from melanoma after treatment with high ROS showed a more invasive phenotype, which may be related to ROS-dependent TNF-α secretion [117]. The removal of ROS through the STAT3 signaling pathway can selectively inhibit the polarization of M2 macrophages to enhance the antitumor function [118]. Coincidentally, high expression of PD-L1 has been found to be associated with M2 polarization in macrophages [119]. Moreover, macrophages acquire an immunosuppressive phenotype and increase the expression of PD-L1 when treated with ROS inducers, such as the glutathione synthesis inhibitor and paclitaxel. Mechanistically, the accumulation of ROS activates NF-κB signaling to promote the transcription of PD-L1 and the release of immunosuppressive chemokines [120]. Therefore, ROS may regulate PD-L1 expression through altering the macrophage M1/M2 balance.

***Regulatory T (Treg) cells.*** Tregs are a subtype of CD4^+^ T lymphocytes expressing the transcription factor forkhead box protein P3 (FOXP3). Emerging evidence has shown that the TME recruits Treg cells to exert its powerful immunosuppressive effects [121]. Generally, low ROS levels are believed to be responsible for the hypofunction of Tregs. Specifically, Treg-mediated direct inhibition of CD4^+^ effector T cells was blocked or reduced when ROS was eliminated by antioxidants or NADPH oxidase inhibitors [122]. Additionally, ROS may promote Treg-cell differentiation [123]. Metformin could inhibit mitochondrial ROS, thus blocking the differentiation of immature CD4^+^ T cells into Treg cells. Then, the expression of FOXP3 was downregulated and ultimately lead to reduced numbers of tumor-infiltrating Treg cells [124]. Conversely, ROS can also maintain the function of Tregs. Specifically, SENP3 is a positive regulator that maintains the stability and promotes the function of Treg cells. For instance, TCR signaling-induced ROS specifically inhibited the degradation of SENP3, thereby maintaining the immunosuppressive property of Treg cells [79]. Treatment with NAC can inhibit SENP3 expression and disrupt the stability of Treg cells, thereby improving the tumor immune response. Intriguingly, compared to effector CD4^+^ T cells, Tregs are less susceptible to cell death induced by oxidative stress. This finding may be ascribed to higher expression levels of antioxidative enzymes in Tregs [125]. In conclusion, ROS is essential in regulating the function of Tregs. The immunosuppression of Tregs seems to be closely associated with ROS levels.

***Myeloid-derived suppressor cells.*** MDSCs are heterogeneous cell groups that consist of myeloid progenitor cells and immature bone marrow cells (IMCs), which are precursors of dendritic cells (DCs), macrophages and/or granulocytes with the ability to significantly suppress immune cell responses [126]. Under physiological conditions, MDSCs can differentiate into corresponding mature cells to perform normal immune functions. ROS are essential in maintaining the undifferentiated state of MDSCs. Studies have demonstrated that the reduction in endogenous H_2_O_2_ levels in MDCS can promote their differentiation into antigen-presenting cells [127]. Significantly, in a murine model of colon cancer, H_2_O_2_ scavenging with catalase induces myeloid differentiation into macrophages [128]. Similarly, MDSCs differentiate into DCs and macrophages in the absence of NADPH oxidase (NOX) activity [128]. Therefore, the mechanism of inhibiting the differentiation of MDSCs may be related to endogenous oxidative stress.

#### 3.2.4. Crosstalk between Different Immune Cells

ROS can function as intercellular messengers for MHCII-restricted antigen presentation. T-cell activation may also be mediated by ROS derived from surrounding cells. For instance, DCs have been shown to generate ROS during antigen presentation [129], while the activation of T cells relies on close contacts between APCs, forming a closed compartment called the immune synapse. This structure helps to enhance the affinity of TCR and antigen peptide–MHC complex interactions, thus promoting the interaction between antigens presenting cells and T-cell signaling molecules [130]. Moreover, ROS are one of the mechanisms responsible for MDSCs-mediated suppression of T-cell activation. MDSCs also influence other immune cells through ROS. It has been reported that ROS from MDSCs can inactivate T cells and prevent the initiation of an immune response [131]. It was also reported that immunosuppression could be reversed by scavenging ROS in MDSCs to exert antitumor effects. Furthermore, NK cells are also a key class of effector cells in response to adenovirus infection. However, ROS generated by MDSCs prevent NK cells from responding to adenoviral vectors and vaccinia virus infection [132]. In the light of recent studies, MDSCs can also negatively regulate B-cell-mediated antitumor immune responses primarily through ROS [133,134]

Collectively, these findings suggest that ROS could not only directly affect immune cells, but also regulate the interactions between different types of immune cells, thereby affecting the tumor immune microenvironment. In line with this, a large number of studies have shown that regulating ROS levels in a tumor-immune microenvironment can improve immunotherapeutic efficacy [135,136]. Therefore, different types of immune cells in the immune system may coordinately respond to ROS stimulation, thus affecting the efficacy of tumor immunotherapy.

## 4. Manipulating Oxidative Stress for Cancer Immunotherapy

Despite the promising potential of targeting the immune system for cancer therapy, the clinical application of immunotherapy faces many challenges in terms of efficacy and safety. For instance, CAR-T against CD19 has been proven to be effective in the treatment of acute lymphoblastic leukemia, but some melanoma and non-small-cell lung cancer patients still do not benefit from these treatments [137,138]. Furthermore, immunotherapy can induce serious side effects in some patients, including autoimmunity and nonspecific inflammation. Therefore, there is an urgent need to improve the safety of immunotherapy while ensuring the efficacy. Given that ROS-induced oxidative stress has multifaceted roles in the immune response to affect the efficacy of immunotherapy [139,140], it alerts us to fact that targeting oxidative stress in the tumor microenvironment may potentiate tumor immunotherapy. Among the therapeutic strategies harnessing ROS for improving cancer immunotherapy, nanomaterials, a class of materials with the sizes of 1–100 nm, have exhibited favorable effects [141,142,143]. Furthermore, a number of nanomaterials have been shown to elicit immunogenicity by inducing immunogenic cell death (ICD).

### 4.1. The Impact of ROS on ICD

ICD is a modality of cell death when a variety of chemical drugs and radiation are used to treat tumors. Tumor cells transform from nonimmunogenic cells to immunogenic cells to mediate the immune response [144,145]. The occurrence of ICD involves a series of signaling molecules and cytokines, including changes in the expression levels of signaling molecules on the cell membrane surface and the synthesis and release of DAMPs [146]. Concretely, DAMPs also promote the recruitment, maturation and activation of APC, mediating the tumor antigen delivery to CD8^+^ effector T cells, thus contributing to the selection and activation of antigen-specific T cells [147]. Some proven ICD inducer chemotherapeutics, such as oxaliplatin, bleomycin and doxorubicin, can induce ROS-mediated immunotherapy. Excessive ROS could induce the ICD of tumors, providing a potential antigenic stimulation for the immune system [148]. According to the status of cytotoxic lymphocyte infiltration, solid tumors can be classified as immunologically ‘hot’ (high immunogenicity) or ‘cold’ (low immunogenicity) [149]. Studies suggest that ROS can increase the infiltration of DCs and T cells in the tumor-immune microenvironment and transform cold tumors into hot tumors, thereby increasing a more effective antitumor immune response [150,151,152]. Nevertheless, the elimination of ROS causes an increase in antitumor immunity and T-lymphocyte infiltration, resulting in a potent antitumor effect [153]. A previous study found that, in breast cancer, the removal of ROS in the TME alleviates the immunosuppressive ICD induced by oleandrin anticancer drugs and induces a longer-lasting effect on T cells [153]. Collectively, these contradictory results prompt that ICD can be induced by modulating ROS to obtain enhanced immunotherapy.

### 4.2. ROS-Regulating Immunostimulatory Nanomedicines

Recently, it has been demonstrated that both chemotherapeutics and photodynamic therapy (PDT) trigger ROS generation, which induces ER stress and promotes DAMP release and ICD cascades [154,155,156]. However, chemotherapy causes multiple side effects, such as pain, bone marrow suppression, hair loss and organ damage. Unfortunately, in practical clinical applications, ICD-based immunotherapy is often limited by the insufficient induction of ICD. To improve efficacy and control these adverse effects, nanomedicine holds great potential for enhancing the efficacy of cancer immunotherapy. Nanoparticles increase the accumulation of targeted drugs at the tumor site, enhance cellular uptake and enable more effective targeting of the desired tumor and/or immune cells, finally triggering the release of ICD (Figure 6).

#### 4.2.1. Photodynamic Immunotherapy of Cancers Based on ROS

Photodynamic therapy (PDT) is a noninvasive treatment that takes advantage of ROS created by photosensitizers (PSs) to repress tumors. PSs are a class of compounds that can absorb ultraviolet rays of a certain wavelength to generate free radicals or ions and initiate photopolymerization. The ideal photosensitizer should at least satisfy the following conditions: a high chemical purity, minimal cytotoxicity and high selectivity of tumor tissues [157]. After the development of two generations of photosensitizers, the current third generation of photosensitizers aims to design the targeted portion of PS to improve tissue selectivity. PSs are irradiated and excited by light at a specific wavelength and then transfer energy to the surrounding oxygen to generate ROS. Compared with radiotherapy and chemotherapy, PDT has fewer side effects and systemic toxicity [158]. In general, oxidative stress induced by PDT induces the activation of innate immune cells and the expression of IL-1, IL-6 and TNF-γ. As mentioned above, PDT can induce ICD and release tumor-associated antigens, stimulating the activation and proliferation of CD8^+^ T cells [159]. It has been shown that fluorine-assembled photodynamic immunotherapy (PMPt) generates sufficient ROS upon laser irradiation to release cisplatin-conjugated PMPt, which inhibits tumor growth by destroying Treg cells and MDSCs [150]. Moreover, PDT has been tested in animal models combined with different immunotherapeutic modalities. Several recent preclinical findings reported in preclinical studies demonstrated the beneficial antitumor effects of ICIs in combination with PDT [160]. In addition, a self-assembled nanoscale coordination polymer (NCP) nanoparticle loaded with oxaliplatin was designed and coated with a layer of photosensitizer pyrolipid on the outer layer. The experimental results show that, in the CT26 bilateral tumor model, the use of PD-L1 blockade and the generation of large amounts of ROS by light induced increased CD8^+^ T-cell infiltration in the primary and distal tumors, and eventually the regression of both tumors could be observed [161]. However, an inadequate oxygen supply or hypoxia is a common feature of solid tumors, resulting in reduced photosensitizer-induced ROS generation, which greatly limits the efficacy of PDTs in clinical treatment. To overcome this problem, a large number of studies have focused on in situ generation of oxygen or delivery of oxygen to the tumor through catalytic reactions [162,163]. One study designed a novel catalytic nanoplatform (nGO-hemin-Ce6). Compared to the control nanosystem without co-encapsulating hemin (a catalase-mimetic nanozyme), the nanosystem has a better tumor-suppression effect [164].

#### 4.2.2. Metallic Immunotherapy of Cancers Based on ROS

Metal nanoparticles (MNPs) with a smaller size and higher tensile strength can penetrate deep tumors together with target materials [165]. Many metal nanoparticles generate ROS by interacting with mitochondria in cells. Nanoparticles can be helpful to reduce the transfer of electrons and process the mitochondrial membrane potential, leading to the excessive accumulation of ROS. Various metal nanoparticles (such as Fe2^+^, Cu2^+^ and Mn4^+^) can generate large amounts of ROS through Fenton and Fenton-like reactions [166,167,168,169]. For example, iron oxide generates ROS by reacting with intracellular H_2_O_2_ via the Fenton reaction. The increased levels of ROS induce the secretion of proinflammatory cytokines at both the tumor site and in the systemic circulation, thereby improving the immune response. Another study has shown that magnetite IONPs polarize M2 macrophages to M1 macrophages by downregulating the expression of M2-associated arginase 1 (Arg-1) [170]. Similarly, in the CT26 tumor model, animals treated with gold nanospheres containing pardaxin (FAL) peptide and indocyanine green (ICG) (FAL-ICG-HAuNSs) were found to produce significantly higher levels of ROS than the untreated control group. It was found that the combined treatments of PDT and photothermal therapy (PTT) exhibited a synergistic antitumor effect [171]. Another study designed a core–shell nanoparticle (UCSs) combining a metal–organic framework (MOF) as the shell for the combinational therapy against hypoxic tumors [172]. In addition, improving the structure and composition of nMOFs is expected to create a new clinically applicable nanotechnology platform to facilitate the immune stimulation of the tumor microenvironment, deliver cancer vaccines, mediate catalytic PDT and ICD, and improve cancer immunotherapy [173].

#### 4.2.3. Other Immunotherapy of Cancers Based on ROS

Cancer cells frequently face metabolic stress in the tumor microenvironment [174], which may lead to excessive intracellular ROS production. Given the role of ROS in regulating cancer immunotherapy, targeting metabolic pathways would be a promising strategy to modulate immunotherapeutic efficacy. The metabolic reprograming of cancer cells makes them highly dependent on certain nutrients, such as glucose and glutamine [175]. In cancer cells, glucose metabolism pathways include glycolysis, the pentose phosphate pathway (PPP) and the tricarboxylic acid cycle. According to the Warburg effect, cancer cells commonly use glycolysis to generate energy, even in an aerobic environment [176]. Moreover, T-cell activation and differentiation are largely dependent on glucose metabolism through the glycolytic pathway [177]. In summary, these findings suggest that targeting glucose metabolism may become an effective strategy to manipulate ROS for enhancing immunotherapy [178].

## 5. Conclusions and Perspectives

As signaling molecules or cellular toxicants, ROS are continuously generated, transformed and consumed in a variety of biochemical reactions. The traditional view links ROS to inflammation, aging and cancer. Emerging data show that ROS play important roles in immune responses. Practically, most immune cells are affected by the concentrations of ROS and thus cannot normally exert their antitumor effect in the tumor microenvironment, resulting in the resistance to some immunotherapy. Nanotechnology facilitates the development of multimodal immunotherapy strategies for cancer treatment. For example, immunotherapy based on the ROS-producing nanomaterial PDT simultaneously enhances immunity against primary and metastatic tumors [179]. In the future, immunotherapy combined with a ROS-targeting strategy based on nanotechnology seems to be a more promising and favorable anti-tumor strategy. However, with few clinical experimental data, future studies need to focus on the safety of nanomaterials. Meanwhile, it is an interesting hypothesis that ROS may transduce across membranes to affect the function of neighboring immune cells. However, there are few studies, to date, focusing on ROS shuttling in the tumor microenvironment and the subsequent effect on immunotherapy. Some types of ROS are prone to permeabilize plasma membranes, such as H_2_O_2_ [180,181]. Nevertheless, these kinds of ROS are usually unstable and easily decomposed in the microenvironment. Additionally, the relevant methods for detecting ROS in the tumor microenvironment are still limited. In this regard, the biological function and underlying mechanisms of ROS transduction across different cells remain to be further explored.

## Figures and Tables

**Figure 1 antioxidants-11-00853-f001:**
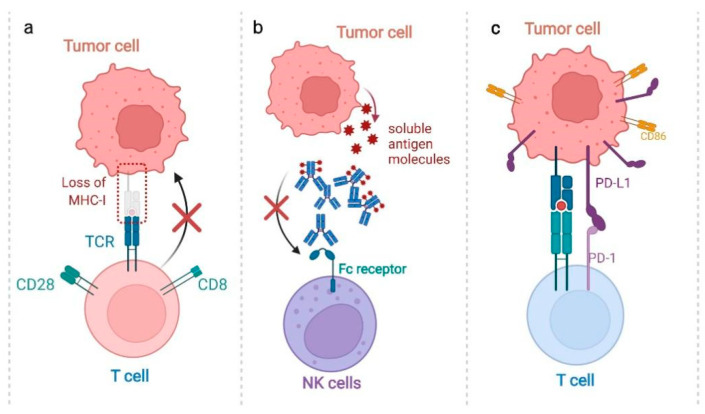
The mechanisms underlying tumor immune escape. (**a**) Low immunogenicity: Cancer cells can escape immune surveillance through downregulating the tumor-specific antigen. Cancer cells with strong immunogenicity induce effective antitumor immune responses and are easily eliminated, while cells with negative antigen and downregulated MHC-I/II expression can escape from the immune system. (**b**) Inhibited ADCC: Cancer cells could also release antigen molecules, which form complexes with antitumor antibodies. Free antibodies could bind to Fc receptors through the Fc segment of the antibody, and then mediate antibody-dependent cellular cytotoxicity (ADCC). However, the depletion of antitumor antibodies by soluble antigen molecules and the blocking of Fc receptors inhibits ADCC and promote tumor immune escape. (**c**) Overexpressed immune checkpoint: Cancer cells are capable of escaping from the immune surveillance by overexpressing B7 family molecules (such as CD86 or PD-L1), which suppress antitumor T-cell responses by binding to the PD-1 receptor.

**Figure 2 antioxidants-11-00853-f002:**
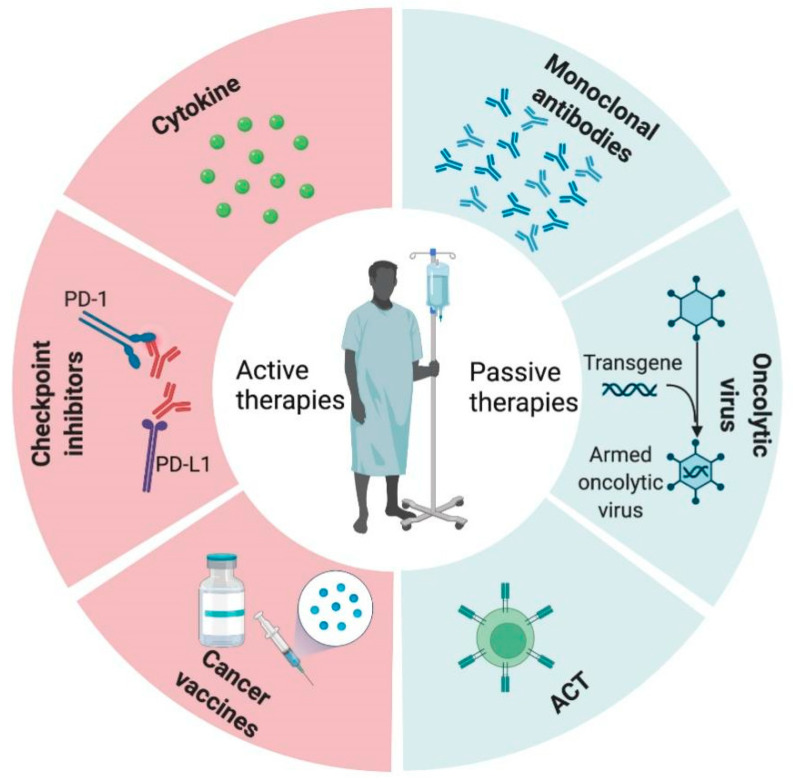
Classification of tumor immunotherapy. Immunotherapy is categorized as either ‘active’ (checkpoint inhibitors, cytokine and cancer vaccines) or ‘passive’ (monoclonal antibodies, oncolytic viruses and ACT).

**Figure 3 antioxidants-11-00853-f003:**
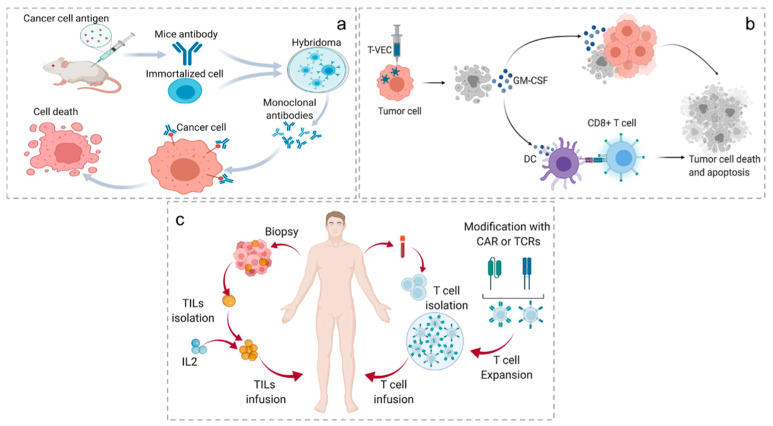
Mechanisms of passive immunotherapy. (**a**) Immunization of mice with cancer-specific antigens to stimulate antibody production, antibody–cell fusion, hybridoma selection and amplification. Then, monoclonal antibody binds to cancer cells. Finally, effector immune cells recognize antibodies bound to cancer cells, leading to cell death. (**b**) After the patient was injected with engineered T-VECs, apoptotic tumor cells released GM-CSF. The specific effects are as follows: local effect (the released GM-CSF causes the apoptosis of surrounding tumor cells) and systemic effect (GM-CSF activates DCs, activating CD8^+^ T cells). Finally, tumor cell death and apoptosis occur. (**c**) CTLs are collected from solid tumors and expanded ex vivo with the systemic administration of IL-2. T cells are acquired from the blood of patients, and then CAR/TCR T-cell chimeric antigen receptors are created. After extensive amplification, they are reinjected into the patient.

**Figure 4 antioxidants-11-00853-f004:**
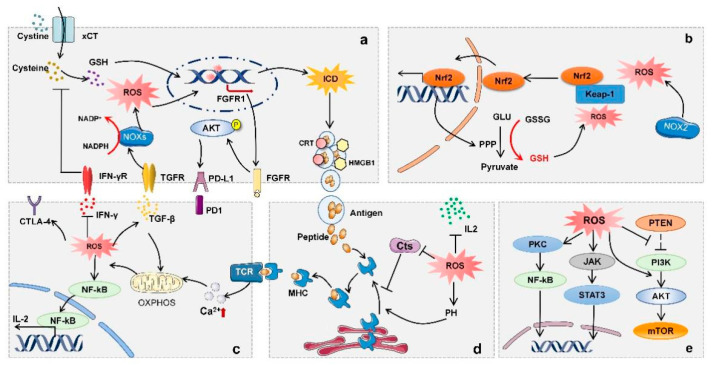
The role of ROS in regulating immune cells. (**a**) Tumor cells: IFN-γ signaling downregulates the expression of xCT, thereby decreasing the synthesis of GSH. TGF-β signaling activates NADPH oxidase to increase ROS production. Oxidative damage ultimately leads to cancer cell death and induces ICD, resulting in significant production of tumor neoantigens. In addition, ROS can promote the expression of FGFR1 to activate the Akt signaling pathway, contributing to a significant increase in PD-L1 protein. (**b**) MDSCs: ROS generated by NOX2 induce a conformational change in Keap1 that releases nuclear factor erythroid 2-related factor 2 (Nrf2). The activation of Nrf2 leads to the increased expression of GSH-transmitting PPP, and, subsequently, increased glutathione can affect MDSCs differentiation. (**c**) The stimulation of TCR induces the release of intracellular calcium ions to activate OXPHOS. ROS generated through OXPHOS then activate NF-kB and induce IL-2 expression, leading to enhanced T-cell proliferation. (**d**) DCs: ROS can affect antigen processing by regulating pH and inhibiting Cts (performing proteolytic activity at a specified pH), thereby affecting the binding of antigens to MHC molecules. (**e**) B cells: ROS participates in BCR signaling pathways, including PKC/NF-κB and PI3KA/mTOR.

**Figure 5 antioxidants-11-00853-f005:**
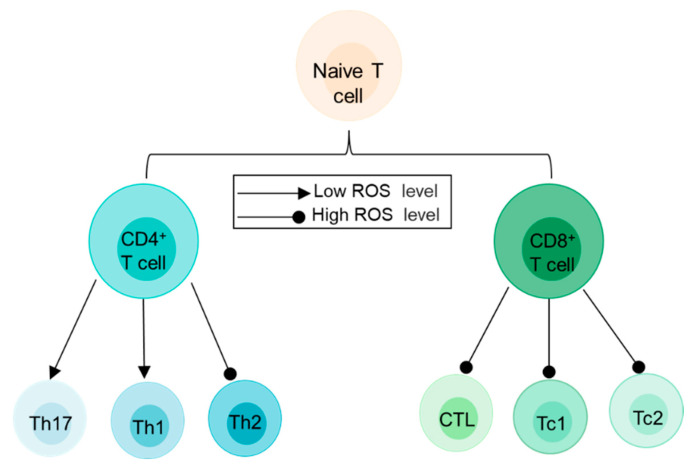
ROS play different roles in T-cell differentiation of different subpopulations. In naive CD4^+^ T cells, low levels of ROS may promote both Th1 and Th17 cell differentiation, while high levels of ROS promote the development of Th2 cells. After activation, naïve CD8^+^ T cells differentiate into mature CTL, Tc1 or Tc2 cells, when experiencing high levels of ROS.

**Figure 6 antioxidants-11-00853-f006:**
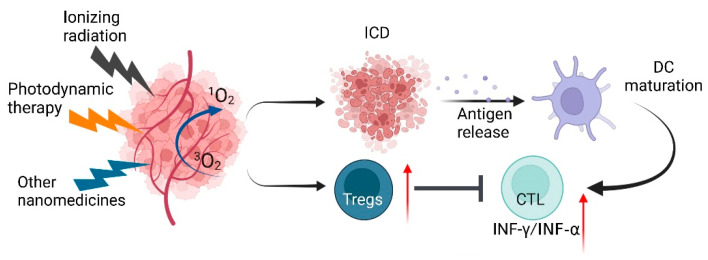
Nanomedicine-induced ICD. Schematic illustration showing that the various nanoparticles (ionizing radiation, photodynamic and other nanomedicines) induce tumor cells to produce a large amount of ROS, thereby inducing ICD and upregulating Tregs. ICD can release many tumor neoantigens, which are phagocytosed by dendritic cells (DCs). DCs mature after cytokine stimulation and present antigens to T lymphocytes. Activated T lymphocytes become effector T cells and are attracted by chemokines, and then migrate to tumors and kill tumor cells. ROS: Reactive oxygen species; ICD: immunogenic cell death; and DCs: dendritic cells.

**Table 1 antioxidants-11-00853-t001:** Summary of ROS effects in different subpopulations of T cells.

Classification Criteria	Subpopulations of T Cells	Effects of ROS	Produce ROS or Not	References
Subtype of TCR	γδT cell	Differentiation	Yes	[71]
	αδT cell	Differentiation	Yes	[71]
Subtype of CD	CD4^+^ T cell	Activation Differentiation	Yes	[72,73,74]
	CD8^+^ T cell	Activation Differentiation	Yes	[75,76]
Function	Th cell	Differentiation	Yes	[77,78]
	Tc cell	Activation	Yes	[74]
	Tregs	Accumulation in the oxidative microenvironment Tregs-mediated suppression	Yes	[79,80]
Activation phase	Naive T cells	Activation Differentiation	Yes	[74,81]
	Effector T cell	Activation Differentiation	Yes	[82]
	Memory T cell	Formation Maintenance	Yes	[83,84]

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
