# Peer review of "Oxidative Stress in Cancer Immunotherapy: Molecular Mechanisms and Potential Applications"

_antioxidants, 2022, doi:10.3390/antiox11050853_

Round 1

Reviewer 1 Report

The manuscript by Liu et al. is well-written and can be a good review for researchers who are new to regulation of ROS in immunotherapy. Here're few suggestions I would like to provide to enhance the significance of this manuscript:

1. May the authors provide a table that includes all the essential abbreviation? It will be helpful for readers to check the original form of the words when needed.

2. Line 102: May the authors add "definition" for active/passive therapies? 

3. Line 159: The oncolytic virus could be a interesting point for immunotherapies. I wonder if the authors can list some example for natural occurred OVs observed in tumors.

4. Figure.3: May the authors briefly describe what the GM-CSF is? The functions and its role in ROS regulation could be included.

5. Line231: The authors mentioned a few antioxidant systems, but lacking explanation and description.

6. Line251: Please check the reference for inflammatory cytokines.

7. Figure4d: May the authors include description of "Cts" in both figure legend and the manuscript?

8. It seems level and functions of ROS varies at different stages of T cell differentiation. The level and signaling pathway could also guide the fate of T cells. May the authors provide a T cell differentiation hierarchy figures showing the timing and fate decision potentially mediated by ROS?

9. In addition, there're many subpopulation of T cells. The authors can add up a table to list the effects of ROS in different T cells. Moreover, the authors could add a column showing if each T cell subpopulation produce ROS or not.

10. Line386: May the authors make some examples how ROS is enhanced in TAM?

11. Line425-429: It seems the description provide inconsistent information. Please check whether H2O2 uptake induces or reduced the differentiation of immature myeloid cells into macrophages or not.

12. Line505: May the authors provide more detail information about the photosensitizers? Like the authors mentioned, the PS can be a breakthrough for immunotherapy. The authors could describe more about its chemical characteristics and how it's designed.

13. Although we have gained many knowledge about the effects and potential roles of ROS in each individual immune cells, it remains unclear how the immune system balances ROS to control the activation of immune cells against cancer cells. On the other hand, I wonder how the authors think the transduction and sensor of ROS across different cells. Some ROS are prone to permeabilize plasma membranes, some are not.

May the authors provide some insights about these point?  

Reviewer 2 Report

The manuscript by Liu et al (i) provides a brief description of the mechanisms used by tumour cells to escape the immune response, (ii) overviews the main immunotherapeutic strategies, (iii) discusses the role of oxidative stress (and ROS) in the immune response and in affecting the success of immunotherapies, and (iv) highlights opportunities to manipulate oxidative stress to increase the success of immunotherapies.

This review deals with a very timely issue (ie, the efficacy of immunotherapies), and the authors provide a good overview of the multifaceted (and sometimes opposite) roles of ROS in the immune response.

The role of the immune system in suppressing tumour development (and the ways in which tumours escape the immune system) as well as the role of ROS as either signaling or damaging molecules are very complex topics. Regarding how ROS can be manipulated to improve the success of immunotherapies, the focus of this review is on nanotechnologies/nanoparticles. It would be good to mention if this is the only way that ROS can be manipulated to modulate the response to immunotherapies. On lines 486-487 it is stated that “Currently, it has been demonstrated that both chemotherapeutics and photodynamic therapy (PDT) trigger ROS generation…”, but the ensuing discussion is focused on PDT and metallic immunotherapies. Are there are any other ways (proposed, explored or envisioned) that ROS can be manipulated to increase the efficacy of immunotherapies? For instance, glucose starvation can induce oxidative stress – can the success of immunotherapies increased by nutrient deprivation or other ways to alter the metabolism of cancer cells into affecting the levels of ROS?    

Minor editorial suggestions/typos:

  • “In detail,” – change to “Specifically,”
  • Lines 415-416 and 417-418 are identical
  • Lines 419-420: “are heterogeneous cell groups that consist of myeloid” is repeated
  • Line 470: “This substance is known as DAMPs” – rephrase
  • Line 480: “Currently” – change to “Recently,”
  • Figure 5: “N” is missing in “Other anomedicines”
  • Line 511-516: “However, inadequate oxygen supply or hypoxia is a common feature of solid tumors, resulting in reduced ROS production, which greatly limits the efficacy of PDT in clinical treatment. It has been shown that fluorine-assembled photodynamic immunotherapy (PMPt) generates sufficient ROS upon laser irradiation to release cisplatin-conjugated PMPt, which inhibits tumor growth by destroying Treg cells and MDSCs [130].” Not clear how/if the two statements are connected (it feels as if the second statement should address the issue in the first)…
  • Line 517: “recently findings” – change to “recent findings”
  • Line 251 and 527: replace “ref” with the citations

Reviewer 3 Report

In this review Liu and colleagues covered the connection between ROS function and immune response, and the potential therapeutic window of the modulation of ROS biology that could enhance immunotherapy efficacy.  

This is a well written and elegantly presented review, which raises interesting scientific conclusions with potential clinical impact.  I have major and minor concerns that in my opinion will improve the quality of this review.

Major: 

The introduction (lines 32 -98) is very general, and does not contribute much conceptually to the review. The article could start with the overview of cancer immunotherapy directly.

Minor: 

1. Missing references in lines 251 and 527. 

2. Figure 3A, check the fonts and cartoons resolution. 

3. Check spelling Figure 5.

Round 2

Reviewer 1 Report

I am satisfied with the authors' responses. 

Author Response

Thank you for reviewing our manuscript and for the constructive comments, which greatly helped us to improve the manuscript.

Reviewer 2 Report

In this revised version, the authors addressed adequately my concerns/suggestions. Overall, the manuscript is also greatly improved through the addition of new information, new points, better explanation of some statements. Below I am listing some minor edits associated with the new material added to this version.

  • Line 156: “has a natural selection for cancer cells”; “natural selection” is not the best way to describe this… do you mean “natural preference”?
  • Line 161: “to insert”- change to “to include”
  • Line 164-65: “to improving the anti-tumor and anti-infection capabilities” – change to “to improve their anti-tumor and anti-infection capabilities”
  • Line 165: “ROS is closely related to Crohn's disease (CD)” – rephrase
  • Line 167: “with CD that GM-CSF” – change to “with CD reported that GM-CSF”
  • Line 169: “change OVs” to “OV”
  • Line 232” In addition” – change to “Furthermore” or “Also”, to avoid having two sentences starting with “In addition”
  • Lien 297: “subpopulation” change to “subpopulations”
  • Table 1: not clear what “expressed of” means
  • Line 334: “cell differentiation for different subpopulation” – change to “cell differentiation of different subpopulations”
  • Line 585: add a reference after the statement “T cell activation and differentiation are largely dependent on glucose metabolism through the glycolytic pathway”
  •  

Reviewer 3 Report

Dear Authors,

Thank you for addressing the comments. The new version of the manuscript  is clearer and more precise.  

Author Response

(The authors gave the same response as above.)

Round 3

Reviewer 2 Report

The authors addressed all my suggestions.